# Intra-Abdominal Hypertension and Compartment Syndrome after Pediatric Liver Transplantation: Incidence, Risk Factors and Outcome

**DOI:** 10.3390/children9121993

**Published:** 2022-12-18

**Authors:** Norman Junge, Annika Artmann, Nicolas Richter, Florian W. R. Vondran, Dietmar Böthig, Michael Sasse, Harald Köditz, Ulrich Baumann, Philipp Beerbaum, Torsten Kaussen

**Affiliations:** 1Division for Pediatric Gastroenterology and Hepatology, Department of Pediatric Kidney, Liver and Metabolic Diseases, Hannover Medical School, 30625 Hannover, Germany; 2Department of Pediatric Cardiology and Intensive Medicine, Hannover Medical School, 30625 Hannover, Germany; 3Department of General, Visceral and Transplant Surgery, Hannover Medical School, 30625 Hannover, Germany

**Keywords:** abdominal perfusion pressure, staged abdominal wall closure, near-infrared spectroscopy, cold ischemia time, aortal hepatic artery anastomosis

## Abstract

In pediatric liver transplantation (pLT), the risk for the manifestation and relevance of intra-abdominal hypertension (IAH) and abdominal compartment syndrome (ACS) is high. This observational study aimed to evaluate the incidence, relevance and risk factors for IAH and ACS by monitoring the intra-abdominal pressure (IAP), macro- and microcirculation (near-infrared spectroscopy (NIRS)), clinical and laboratory status and outcomes of 27 patients (16 female) after pLT (median age at pLT 35 months). Of the patients, 85% developed an elevated IAP, most of them mild. However, 17% achieved IAH° 3, 13% achieved IAH° 4 and 63% developed ACS. A multiple linear regression analysis identified aortal hepatic artery anastomosis and cold ischemia time (CIT) as risk factors for increased IAP and longer CIT and staged abdominal wall closure for ACS. ACS patients had significantly longer mechanical ventilation (*p* = 0.004) and LOS-PICU (*p* = 0.003). No significant correlation between NIRS or biliary complications and IAH or ACS could be shown. IAH and ACS after pLT were frequent. NIRS or grade of IAH alone should not be used for monitoring. A longer CIT is an important risk factor for higher IAP and ACS. Therefore, approaches such as the ex vivo machine perfusion of donor organs, reducing CIT effects on them, have great potential. Our study provides important basics for studying such approaches.

## 1. Introduction

Pediatric liver transplantation (pLT) is still a challenging procedure with excellent long-term outcomes and a 5-year patient survival rate of 97.5% [1]. However, graft survival is still worse (5-year graft survival rate: 86.5%) [1]. The most critical periods include the immediate perioperative phase and the first year after pLT [2]. During these periods, intra-abdominal hypertension (IAH) and abdominal compartment syndrome (ACS) are likely to play important roles given that pLT is at a high risk of developing IAH/ACS and grafts are vulnerable to IAH/ACS, with short-term (thrombosis) and mid-term (ischemic type biliary lesions (ITBL)) consequences.

To date, little attention has been paid to the problem of postoperative increased intra-abdominal pressure (IAP), which can occur after pLT, for example, as a result of transplant size mismatch (“large for size” pLT) or accompanying systemic (hyper) inflammation. In 2013, Biancofiore was one of the first authors to identify IAH as a risk factor in adults undergoing LT [3,4], and until now, only two pediatric studies have investigated IAH in the setting of LT [5,6]. Despite this pioneering work, many questions remain unanswered in both adult and pediatric transplantations and require clarifying studies.

The 2013 World Society of Abdominal Compartment Syndrome (WSACS) [7] defines IAH in children as an IAP of more than 10 mmHg. Liang et al. [8] showed that IAP was a good predictor (area under the curve (AUC) 0.74; cut-off 12.1 mmHg) of 28-day mortality for their pediatric intensive care unit (PICU) cohort. As soon as an IAH is accompanied by new or aggravated organ dysfunction, by definition, it becomes an ACS, which is associated with a mortality rate of well over 60% [9,10,11,12].

The level of IAP elevation does not necessarily correlate with the development or severity of organ failure [13]. However, an association between IAH-related perfusion restriction and histopathological tissue damage in organs could be shown in animal models [14], suggesting that IAP-related macro- and microcirculation changes could be suitable early markers for IAH-related organ damage. Animal and human studies have shown that in cases of IAH, microcirculation becomes impaired in abdominal organs, especially in the liver and kidneys [15]. This underlines the assumption that through changes in microcirculation, IAH could have significant impacts on liver grafts and the clinical course after pLT.

Various measurement methods for the quantification of microcirculation have been tested in pediatrics [16], but an application in IAH or ACS has not yet been established. In our study, we use near-infrared spectroscopy (NIRS), which detects transcutaneous, regional oxygen saturation as a function of tissue perfusion or equilibrium from supply and demand, and duplex sonography-based dynamic tissue perfusion measurement (DTPM, PixelFlux) analyses. The latter was first described by Scholbach et al. [17,18].

The main aims of our study were to evaluate the incidence, risk factors and impacts on the outcomes of IAH and ACS in patients with pLT. The secondary aims were to evaluate the microcirculation markers for IAH and ACS and the association of IAH severity and ACS. The long-term goal was to identify reliable monitoring markers and addressable risk factors for IAH and ACS, which could further improve the prognosis of pLT.

## 2. Materials and Methods

Patients and ethics

This observational study included patients after pLT at the Children’s Hospital of Hannover Medical School from January 2015 to April 2016. The follow-up time ended in June 2019. Informed and written consent was obtained from the patients’ guardians. Exclusion criteria were congenital malformations and conditions after surgery or injury of the urogenital or upper gastrointestinal tract leading to contraindications for intragastric or intravesical pressure measurement. Esophageal varices were not exclusion criteria. Immunosuppressive protocols are shown in the Appendix A. On 14 January 2016, the study was approved by the local ethics committee (MHH-no. 6677). Data protection regulations were fulfilled, and the Declaration of Helsinki principles were followed. The study was registered (WHO-ICTRP-ID: DRKS00006556).

Data collection

Detailed clinical status, extensive blood values (inflammation marker, blood count, coagulation, blood gas analysis, liver and kidney parameter), IAP and microcirculation were monitored for 5 days after pLT; graft/patient survival and laboratory status in median over 4.1 years.

Patient, transplant-procedure and graft characteristics

The following parameters were analyzed: living (LD-pLT) versus deceased donor (DD-pLT), graft and explanted liver weight ratio, graft size (full-size or split liver), cold ischemia time (CIT), primary (PAWC) versus sequential abdominal wall closure (SAWC) (which means the abdominal wall was closed temporarily without connecting the wound edges), type of hepatic artery anastomosis (HAA), patient’s liver–to–graft liver weight ratio (PL-GL-WR) and graft-to-recipient weight ratio (GRWR). A GRWR of ≥ 4% was considered “large for size” (LFS), and a GRWR of < 4% was considered “acceptable for size” (AFS) [19].

Clinical and laboratory data

Vital signs were recorded continuously, and laboratory parameters were recorded at least every 24 h. Liver function was reflected by minimal coagulation function (quick value), albumin concentration and the total requirement of albumin substitution. Respiratory function was recorded on the basis of the need for mechanical ventilation.

The cardiovascular system was represented by heart rate (HR/min), mean arterial pressure (MAP) and need for catecholamine therapy (mirrored by the vasoactive-inotropic score (VIS) [20], each calculated at midnight and at the time of maximum demand). Analogous to the VIS for catecholamine, renal function was analyzed using a diuresis score that takes into account the relative influence of different diuretics on hourly diuresis performance (see Appendix A).

To compare disease severity, PRISM III scores [21] were calculated on PICU admission and discharge. Hemodynamic, hepatic, renal and respiratory parameters were also monitored.

Intra-abdominal pressure measurement

We monitored IAP via intragastric pressure (IGP) and intravesical pressure (IVP) measurements. For IVP, a modified technique was used according to Kron et al. [22].

IGP was continuously measured using a commercially available system (Spiegelberg, Hamburg, Germany), whose methodology and reliability have already been described by us [23] and others [24,25]. IVP measurements were performed at least 12 times daily, and IGP values were documented hourly. IAP was measured in a supine position and only in nonagitated patients.

Each IAP value is a mean of the corresponding IGP and IVP values. According to WSACS recommendations, only repeatedly elevated values (in sequential measurements) have been interpreted as IAH (IAP > 10 mmHg).

In children, IAH can be divided into the following grades: I = IAP 10–12, II = 13–15, III = 16–18 and IV > 18 mmHg [7,26]. Abdominal perfusion pressure (APP) was calculated by subtracting the IAP from the MAP. ACS was diagnosed in patients with IAH (IAP > 10 mmHg) and new or aggravated organ dysfunction based on International Pediatric Sepsis Consensus Conference (IPSCC) criteria [27].

Abdominal ultrasound

Ultrasound was performed daily on all patients for the first 5 days after pLT, using CompactXtreme “CX-50”, Koninklijke Philips (NV, Eindhoven, Netherlands). The color-Doppler examination was performed using a fixed algorithm, predefined machine setting and standardized projections according to the recommendations of the German Society for Ultrasound in Medicine [28]. Duplex sonographic loops of at least 3 s were stored.

Outcome measurement

The duration of mechanical ventilation (MV), length of stay in the PICU (LOS-PICU) and in the hospital (LOS-hosp), graft and patient survival and the incidence of biliary complications were analyzed for short-term (5 days post-pLT), mid-term (30 days post-pLT) and long-term (median 4.1 years post-pLT) outcomes.

Microcirculation

NIRS (iNVOS5100C, Medtronic, Minneapolis, USA) was applied according to the manufacturer’s instructions. NIRS monitoring was carried out at the thoracolumbar junction at the level of Th12 in projection on the liver. All measurements were subject to a systematic quality and standardized plausibility check.

The previously described standardized recordings of color-Doppler sonographic videos were transferred to a personal computer where DTPM was carried out using PXFX (PixelFlux, Chameleon Enterprises, Münster, Germany). All videos were automatically calibrated for distances and color hues, and a blinded evaluator controlled all loops before analysis. After defining the region of interest (ROI), PXFX automatically carried out the perfusion measurement. An average velocity (v) (cm/s) was calculated for each individual pixel of the ROI area. Perfusion intensity was calculated according to PI (cm/s) = v (cm/s) × A (cm^2^)/A ROI (cm^2^), where A represents area.

Data processing and statistical analysis

Data were collected using an electronic patient data management system (COPRA System GmbH, Berlin, Germany) and then transferred as anonymous data to an Excel database (Microsoft, Redmond, Washington, USA).

A statistical analysis was performed using IBM SPSSv25 Statistics (IBM, Armonk, USA). Data were presented as median, mean ± standard deviation. A *t*-test was applied, if a Kolmogorov–Smirnov and Shapiro test showed normal distribution (IAP, APP, MAP), otherwise a Kruskal–Wallis test (three or more independent groups) and a Mann–Whitney test (two independent groups) were applied. The Chi-squared test was used for the categorical analysis of subgroups.

To conduct a multivariable analysis of factors associated with the development of ACS, we applied binary logistic regression with the following variables: PAWC vs. SAWC, type of HAA (aortal vs. nonaortal), PL-GL-WR and CIT per hour. Linearity was assessed using the Box Tidwell procedure. The presence of multicollinearity was assessed on the basis of a correlation matrix, tolerance and a variance inflation factor. For the analysis of factors associated with higher IAP or lower APP, multiple linear regression with the following variables was applied: CIT (per hour), PAWC vs. SAWC, type of HAA (nonaortal vs. aortal) and PL-GL-WR.

## 3. Results

### 3.1. Study Cohort

We analyzed 27 children (16 girls, median age at pLT 35 months, 95% CI 44–106 months, 0–5 years n = 17; 6–18 years n = 10). In one patient, a combined liver/lung transplantation and in two a re-pLT were performed. Seven patients received LD-pLT and 20 DD-pLT. A left lateral split (segments II + III) was used in 16 patients, a reduced size split in 2 and a full-size liver in 9 patients. Our cohort was divided, according to the presence of IAH and ACS, into three groups: a non-IAH, an IAH and an ACS group. The age at pLT, pediatric risk of mortality score (PRISM score), gender and body weight were independently distributed over the groups (Table 1). Underlying indications for pLT were biliary atresia (n = 12), metabolic illnesses (n = 8; 3× citrullinemia, 2× cystic fibrosis, 2× Wilson disease, 1× Crigler–Najjar syndrome), acute liver failure due to death cap intoxication (n = 2) and other (n = 5; 2× progressive intrahepatic cholestasis, 1× hepatoblastoma, 1× autoimmune sclerosing cholestasis and 1× chronic liver failure after gestational alloimmune liver disease). In 22% of patients (n = 6), there was a size mismatch (GRWR > 4%), all of which were in the age group of 0–5 years. SAWC was performed significantly more frequently in the 0–5 age group (17/20 patients) than in the 6-18 age group (3/7 patients) (*p* < 0.001). On average, in patients with SAWC, the abdominal wall was closed on day 2 (1–7) after pLT.

### 3.2. Intra-abdominal Pressure and Abdominal Perfusion Pressure

The measurement of IAP over the first 5 days after pLT, or until discharge from PICU, showed mean values per day for the entire cohort of below 10 mmHg (Figure 1a). The mean APP for all measurements was 68.3 mmHg (SD ± 12.0; 95% CI 63.6–73.1 mmHg). The day-by-day results are shown in Figure 1b. APP differed significantly depending on age (0–5 years median 59.2, mean 61.9, SD ± 7.9, 95% CI 57.8–65.9 mmHg vs. 6–18 years 76.6, 79.3, SD ± 9.7, 72.4–86.3 mmHg; *p* < 0.001) (Figure 1c); in contrast, IAP did not (*p* = 0.199).

IAP did not differ significantly (*p* = 0.828) in patients with PAWC (mean 8.9; median 8.0; SD ± 2.4; 95% CI 6.6–11.1 mmHg) compared with patients with SAWC (mean 8.7; median 8.9; SD ± 2.0; 95% CI 7.7–9.6 mmHg). The APP for patients with PAWC (mean 82.5; median 85.0; SD ± 9.1; 95% CI 74.05–90.85 mmHg) was significantly higher (*p* < 0.001) compared with patients with SAWC (mean 63.4; median 61.6; SD ± 8.6; 95% CI 59.4–67.4 mmHg). In accordance with this, patients with SAWC had significantly lower MAP (PAWC: mean 92.1, median 92.3, 95% CI 88.0–96.2 mmHg vs. SAWC: 70.9, 67.5, 66.8–75.0 mmHg; *p* < 0.001).

### 3.3. Incidence of Intra-abdominal Hypertension and Acute Compartment Syndrome

Even though the mean values for IAP were below 10 mmHg, 85% of patients developed elevated IAP (most mild to moderate) in at least two consecutive measurements after pLT (Figure 1d). However, 17.4% achieved IAH-grade 3 and 13% IAH-grade 4.

Of the patients, 63% developed an organ dysfunction leading to the diagnosis of ACS in addition to IAH. IAP and APP did not significantly differ between IAH and ACS groups (*p* = 0.114 and *p* = 0.062).

We could not find a significant association between the grade of IAH and that of ACS (distribution shown in Figure 1d). Kidneys were affected most often (Table 2). Mean tacrolimus levels did not diver between patients with and those without renal dysfunction (*p* = 0.520).

### 3.4. Risk Factors for IAH and ACS

The graft and transplant-procedure characteristics for the non-IAH, IAH and ACS groups are shown in Appendix A. The multiple linear regression analysis identified aortal HAA and longer CIT as risk factors for increased IAP (Table 3). SAWC appeared to be a risk factor for reduced APP (Table 4). For ACS, a longer CIT and a longer SAWC could be identified as risk factors (Table 5).

### 3.5. Analysis of Clinical, Laboratory and Vital Parameters, Depending on the Occurrence of IAH or ACS

When comparing the non-IAH, IAH and ACS groups by using a Kruskal–Wallis test, we could not observe any significant differences for heart frequency (*p* = 0.25), mean arterial pressure (*p* = 0.34), respiratory frequency (*p* = 0.39), D-GFR score (*p* = 0.10), albumin blood level (*p* = 0.84) or albumin substitution (*p* = 0.07) but did observe significant differences in the duration of mechanical ventilation (*p* = 0.02) (Appendix A).

### 3.6. Ultrasound Analysis of Liver Perfusion

Depending on IAP levels, and in patients with ACS, portal vein flow was reduced (Figure 2). The resistance index (RI) of the hepatic artery was lower in the IAH group than in the non-IAH group. In contrast, the resistance index of patients in the ACS group was significantly higher than that of the IAH group. The minimum diameter of the retrohepatic vena cava inferior decreased over the groups and was significantly lower in the ACS group than in the IAH group.

### 3.7. IAH/ACS and Outcome

Short-, medium- and long-term patient survival was 100%, 96% and 96%, respectively. One patient died 27 days after pLT, due to pulmonary infection. An outcome analysis on patient survival was not possible, because of limited events. The short-, medium- and long-term graft survival rates were 96%, 89% and 85%, respectively. The development of ACS did not affect graft survival, at either the short-term follow-up or the long-term follow-up (*p* = 0.12).

Outcome characteristics are shown in Table 6. ACS patients had significantly longer mechanical ventilation (*p* = 0.004) and LOS-PICU (Figure 3; *p* = 0.003). The overall hospital LOS around the transplant period was not associated with IAH or ACS. At the end of follow-up, two of four patients (50%) from the non-IAH group showed biliary complications, while 9 of 23 (39%) with elevated IAP (IAH + ACS group together) showed biliary complications. The proportion among patients with ACS was only 18% (3/17). Thus, no significant correlation between biliary complications and previous IAH or ACS could be shown (*p* = 0.10; *p* = 0.67).

Aspartate aminotransferase (AST), alanine aminotransferase (ALT), gamma-glutamyl transferase (GGT), bilirubin, serum cholinesterase (CHE), creatinine, cystatin C, eGFR (based on Schwartz formula) [29] and INR at the last follow-up visit (n = 17 patients, median 4.1 years post-pLT) did not differ significantly between patients with (n = 10) and without ACS (n = 7).

### 3.8. Analysis of Microcirculation Parameters, Depending on the Occurrence of IAH or ACS

When comparing the non-IAH, IAH and ACS groups by using Kruskal–Wallis, we did not observe any significant differences in DTPM (liver *p* = 0.96, kidney *p* = 0.19, spleen *p* = 0.96, intestines *p* = 0.73) or NIRS (*p* = 0.14) (Appendix A).

## 4. Discussion

Here, we present the first prospective study with 27 patients with pLT in whom IAP, ACS criteria, extensive clinical data and liver perfusion by ultrasound and microcirculation were monitored from the first 5 days and the outcome up to 4.5 years after pLT. IAH and ACS seem to be frequent, and risk factors could be identified.

### 4.1. IAP, APP, IAH, ACS and Their Risk Factors in Our Cohort

We detected a high frequency of IAH (85%) and ACS (63%). That is more than reported in an adult study by Freitas (IAH 48%; ACS 15%) [30]. Our non-IAH (n = 4), IAH (n = 6) and ACS (n = 17) groups did not differ in relation to age, gender, body weight or PRISM III score at admission to the PICU. We analyzed IAP by IVP as well as IGP measurements. Continuous IGP measurement is a less complex and less time-consuming method, which was recently validated by our group [23]. In contrast to IGP, IVP is known to underestimate IAP in patients who had liver transplant [31].

IAP was not associated with age or type of abdominal closure in our cohort, which is in line with results published by Deindl et al. [5] We found IAH in 85% (n = 23/27) of our patients, most of whom had mild IAH (1–2°; 69.6%; n = 16/23) (Figure 1). These results are consistent with those of other studies, given that different limits apply to children and adults for anatomical-pathophysiological reasons [32,33]. The lower incidence of IAH and ACS in the Deindl et al. [5] study could be justified by a higher proportion of patients with metabolic diseases (healthier cohort, fewer previous surgeries).

Some studies describe the prognostic value of APP as superior to IAP [34,35,36,37,38]. However, in the pediatric population, APP was not documented in most studies, and its usefulness remains unclear [7,39], especially because of age-dependent differences in physiological MAP. We found that the APP in our patients was within the described critical cut-off for adults (50–60 mmHg [34,40]), but younger patients (age group 0–5 years) showed significantly lower values, which can be explained by a physiologically lower MAP in these patients. Given this background and that SAWC was significantly more frequent in younger patients, it seems obvious why the APP of patients with SAWC was significantly lower than that of patients with PAWC and why SAWC seems to be a risk factor for reduced APP. Therefore, APP should be interpreted with caution in the pediatric population, especially when comparing different age groups.

Of 23 subjects with IAH, 17 developed ACS (74%) (Figure 1). The development of ACS was not associated with any age group. Most patients with ACS suffered renal dysfunction (88%; n = 15/17), and most (n = 13/17, 74%) showed at least one additional organ dysfunction. Renal dysfunction was independent of tacrolimus trough levels in the first five days. Every patient with organ dysfunction compatible with ACS criteria had IAH. However, in contrast to the study by Kathemann et al. [6], the degree of IAP elevation here was not associated with ACS.

Identifying risk factors for IAP elevation or ACS is crucial and cannot be transferred from other diseases/health conditions to patients with pLT. Multiple linear regression revealed CIT and aortal HAA as risk factors for elevated IAP, and binary logistic regression revealed longer CIT and SAWC as risk factors for ACS. The size of donor organs is often only limitedly selectable, and unfortunately, so is CIT. However, organ allocation should be improved to improve size matching and, above all, the length of CIT. Another future approach to reduce the influence of CIT on IAH and ACS could be machine perfusion of donor organs. Regarding the pathomechanism of organ injury due to IAH/ACS, Zhao et al. published an interesting animal study [41] showing that IAH/ACS-induced gut microbiota dysbiosis highly contributes to liver injury. Our study provides vital basics for future investigations into these relevant aspects of the pLT setting.

### 4.2. ACS and PAWC/SAWC

The distribution of ACS among patients with different types of abdominal closure shows that SAWC is not always sufficient in preventing IAH and ACS. In contrast, SAWC was a strong risk factor for ACS. IAP did not differ significantly between SAWC patients and PAWC patients. Therefore, not only is IAP crucial for the development of ACS, but so are other factors, such as systemic inflammatory response syndrome (SIRS) and infections. The similarity of IAP in PAWC patients and SAWC patients must be interpreted with care, as our study was an observational study, and decisions for SAWC versus PAWC were made on a clinical basis. It is possible that patients with SAWC could have developed a much higher IAP if PAWC had been performed. That the association of ACS and SAWC was biased by disease severity is unlikely, as the initial PRISM III score did not differ between SAWC-treated patients and ACS patients.

### 4.3. IAH/ACS and Outcome

As a direct consequence of IAH and ACS, we detected a clear trend toward reduced portal vein flow with a tendency toward zero flow around an IAP of 18 mmHg (Figure 2). This can have a significant impact on outcome by increasing the risk of portal vein thrombosis. In inverse proportion to IAP, decreasing diameters of the inferior vena cava (IVC) were found, which can be interpreted as a clinical sign of compression’s taking place. LOS-PICU was significantly associated with the presence of ACS. However, total hospital stay duration was not associated with ACS.

In the ACS group, three patients suffered from thrombosis after pLT: one patient, who needed revision for portal vein thrombosis on day 2, one patient with the need for a second-look surgery due to ACS and vena cava inferior thrombosis on day 2 and one with portal vein and hepatic artery thrombosis on day 5. In the non-IAH and IAH groups, no thromboses were observed. All patients who needed re-pLT (n = 4) presented with ACS before. However, we could not find any statistically significant impact of IAH/ACS on graft or patient survival, possibly because of the small number of patients. In the long-term outcome data, we could not find any association between IAH/ACS and graft function, kidney function, biliary problems, graft survival or patient survival.

### 4.4. Microcirculation

To the best of our knowledge, we are the first to evaluate microcirculation in relation to IAH and ACS in patients who underwent pLT. IAP measurement is still a stressful procedure for patients and immensely increases the workload for the intensive care unit. NIRS, for example, could be an easier and harmless tool to show the effects of IAH or ACS indirectly, by reduced oxygen saturation in the liver. However, we could not find any association between changes in microcirculation (including NIRS) and the presence of IAH or ACS. This is in contrast to other studies outside the pLT setting [42,43] and the results of our own unpublished studies for patients who had not undergone pLT. In pLT, this could be explained by the extraordinary situation of a denervated liver graft, in combination with hyperdynamic circulation, due to cirrhosis before pLT, leading to an increased total liver blood flow after pLT [44]. This is an important result in that it proves that NIRS is not a reliable marker for IAH/ACS in patients with pLT, even though it has been used in other circumstances and indications. Other studies [45,46] detected an association between NIRS and vascular complications after LT, but this was not within the scope of our study. Another confounder for NIRS measurement in our study could have been the mixed weights and ages of our cohort.

### 4.5. Strengths and Limitations

Because of our study’s prospective nature, extensive data evaluation and long-term follow-up, it is, thus far, unique. The strength of our study is that we classified patients strictly according to international classifications, which only possible was because of the acquisition of copious clinical data.

The main limitation of our study is the small number of patients with, so it has limited statistical power. However, we performed this study to bring awareness to this topic and to create a basis for developing larger, multicenter studies. Another limitation is the heterogeneous nature of our cohort (different diseases and conditions pre-pLT).

## 5. Conclusions

The development of IAH and ACS after pLT was frequent. ACS was associated with IAH, but the IAH grade was not a reliable marker for the occurrence of ACS; likewise, NIRS was not a reliable marker and should not be used for monitoring IAH/ACS. Partly modifiable factors were risk factors for higher IAP (longer CIT, aortal HAA) and ACS (longer CIT, SAWC). In particular, the influence of CIT could be modified by future approaches such as donor organ machine perfusion. Or study provided an important basis for future studies on this topic. ACS increased LOS-PICU and could have an impact on graft survival and thrombosis. However, this needs to be evaluated in further studies with larger cohorts. Our results provided important insights into this topic, creating a basis and highlighting the need for further multicenter studies.

## Figures and Tables

**Figure 1 children-09-01993-f001:**
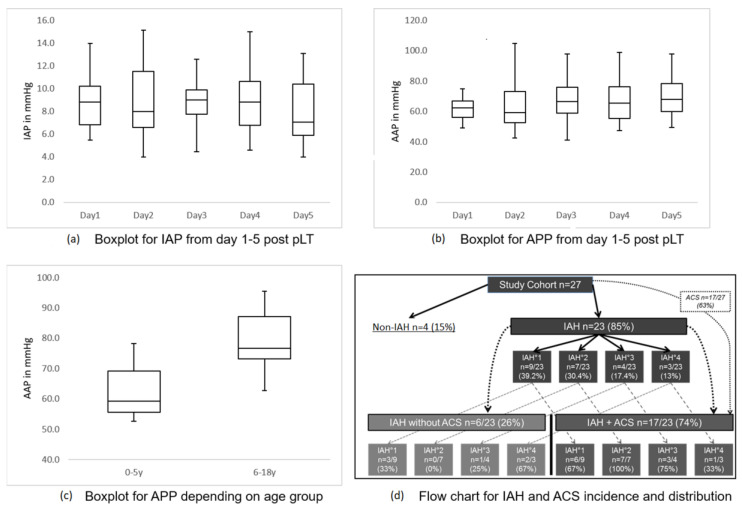
Boxplots for IAP/APP and flow chart for incidences. (**a**) Boxplots for mean intra-abdominal pressure (IAP) in mmHg day per day from day 1 to day 5 after pediatric liver transplantation. (**b**) Boxplots for mean abdominal perfusion pressure (APP) in mmHg day per day from day 1 to day 5 after pediatric liver transplantation. (**c**) Boxplots for mean abdominal perfusion pressure (APP) in mmHg depending on age. (**d**) Flow chart for incidence and distribution of intra-abdominal hypertension (IAH) and abdominal compartment syndrome (ACS) within our study cohort.

**Figure 2 children-09-01993-f002:**
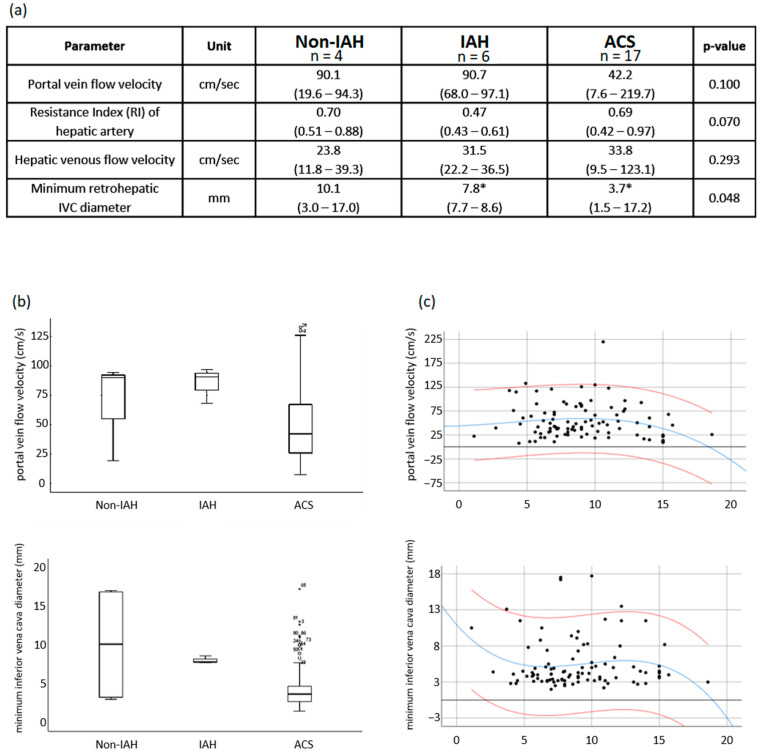
Results of IAP/ACS dependent flow velocities. *(IAH= intra-abdominal hypertension, ACS = abdominal compartment syndrome, IAP = intra-abdominal pressure).* (**a**) Portal and hepatic venous flow velocities resistance index and retrohepatic inferior vena cava (IVC) diameters in 27 children with LT. Determined using duplex sonography; data given as median [min–max]). Nonparametric Kruskal–Wallis test for k independent samples (One-way ANOVA) with post hoc testing according to Dunn-Bonferonni. *: *p* < 0.05. (**b**) Boxplots of portal venous flow velocities and retrohepatic inferior vena cava (IVC) diameters. (**c**) Influence of different IAPs on portal vein flow and retrohepatic IVC diameter, from an IAP of about 15 mmHg IAP-related decreases of flow velocities and/or vein diameter accelerate. In particular, the portal vein flow rate and the IVC width decrease sharply and tend toward 0 cm/s (zeroflow) or 0 mm (complete compression) with an IAP of more than in median 18 mmHg.

**Figure 3 children-09-01993-f003:**
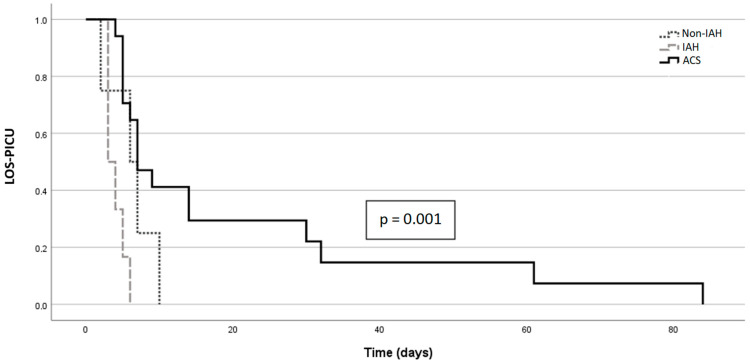
Kaplan-Meier curve for length of stay (in days) in a pediatric intensive care unit (LOS-PICU), depending on having no intra-abdominal hypertension (Non-IAH), having intra-abdominal hypertension (IAH) or having abdominal compartment syndrome (ACS).

**Table 1 children-09-01993-t001:** Patient characteristics.

	Non-IAH	IAH	ACS	Total	Kruskal–Wallis Test *p*-Value
Patients (n =)	4	6	17	27	
Age (years) *	4.00 [±6.68]	8.67 [±7.20]	4.76 [±6.17]	5.52 [±6.45]	0.68
Sex distribution (% girls)	75%	66%	53%	59%	0.67
Weight (kg) *	14.94 [±13.38]	45.02 [±36.41]	21.62 [±21.11]	25.83 [±23.87]	0.62
PRISM III Score * On admission	14.25 [±4.11]	14.83 [±5.19]	18.00 [±25.78]	16.74 [±6.23]	0.45
Previously transplanted	0	0	2	2	0.54

* Mean [standard deviation], PRISM: pediatric risk of mortality score (Crit Care Med. 1996 May; 24(5): 743–52).

**Table 2 children-09-01993-t002:** Distribution of organ dysfunction.

	Total Number in the Study Cohort	Study Cohort in Percentages (n = 27)	All IAH Patients in Percentages (n = 23)	All ACS Patients in Percentages (n = 17)
Renal dysfunction *	15	56%	65%	88%
Cardiovascular dysfunction *	11	41%	48%	65%
Pulmonary dysfunction *	8	30%	35%	47%

* Organ dysfunction was defined by International Pediatric Sepsis Consensus Conference (IPSCC) criteria [27].

**Table 3 children-09-01993-t003:** Characteristics associated with increased IAP in multiple linear regression (n = 27).

Analyzed Parameter/Independent Variable	Coefficient B (Unstandardized)	95% CI	*p*-Value
Cold ischemia time (per hour)	0.26	0.01–0.50	0.044
Abdominal wall closure (primary vs. staged)	−0.65	−2.52–1.22	0.479
Type of arteria hepatic anastomosis (non-aortal vs. aortal)	2.51	0.74–4.28	0.008
Patients’ liver to graft liver weight ratio	1.32	−0.13–2.77	0.073

dependent variable = IAP = intra-abdominal pressure.

**Table 4 children-09-01993-t004:** Characteristics associated with reduced APP in multiple linear regression (n = 27).

Analyzed Parameter/Independent Variable	Coefficient B (Unstandardized)	95% CI	*p*-Value
Cold ischemia time (per hour)	0.89	−0.246–2.01	0.116
Abdominal wall closure (primary vs. staged)	−17.40	−25.83–−8.97	<0.001
Type of arteria hepatic anastomosis (nonaortal vs. aortal)	−5.68	−13.70–2.33	0.156
Patients’ liver–to–graft liver weight ratio	2.63	−3.95–9.21	0.416

dependent variable = APP = abdominal perfusion pressure.

**Table 5 children-09-01993-t005:** Characteristics associated with ACS (odds ratios from binary logistic regression, mutually adjusted for all variables in the model, n = 27).

Analyzed Parameter/Independent Variable	Odds Ratio	95% CI for Odds Ratio	*p*-Value
Abdominal wall closure (primary vs. staged)	35.82	1.64–784.94	0.023
Type of arteria hepatic anastomosis (aortal vs. nonaortal)	1.21	0.10–14.32	0.879
Patients’ liver–to–graft liver weight ratio	1.00	1.00–1.00	0.819
Cold ischemia time per hour	1.57	1.06–2.33	0.023

dependent variable = ACS = abdominal compartment syndrome.

**Table 6 children-09-01993-t006:** Outcome data.

	No IAH	IAH	ACS	Kruskal–Wallis Test *p*-Value	Mann–Whitney U Test for IAH vs. ACS *p*-Value
Patients (n=)	4	6	17		
PRISM III-Score at discharge (mean [±standard deviation])	7.75 [±3.00]	7.50 [±3.21]	8.06 [±7.81]	0.84	
Length of stay in PICU in days (mean [±standard deviation])	6.25 [±3.30]	4.00 [±1.27]	18.94 [±22.5]	0.01	IAH vs. ACS *p* = 0.003
Length of stay in hospital in days (mean [±standard deviation])	35.75 [±14.80]	29.17 [±6.08]	43.53 [+23.87]	0.24	
Length of postoperative invasive ventilation in hours (mean [±standard deviation])	73.75 [±55.14]	17.00 [±11.76]	201 [±307.73]	0.02	IAH vs. ACS *p* = 0.004
Necessity for early retransplantation	0	0	3 (18%)	0.38	
Deceased	0	0	1 (9%)	0.75	

## Data Availability

Data supporting reported results are available on request in anonymized manner.

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
