# Peer review of "Intra-Abdominal Hypertension and Compartment Syndrome after Pediatric Liver Transplantation: Incidence, Risk Factors and Outcome"

_children, 2022, doi:10.3390/children9121993_

Round 1

Reviewer 1 Report

I congratulate the authors to this well-designed prospective study on a highly relevant and interesting topic in pediatric liver transplantation. The authors may discuss a little more in detail possible measures to reduce IAP in pediatric patients after LT. All in all, this is a high-quality prospective study wiht relevance for future research.

Reviewer 2 Report

The manuscript has an ambitious objective, yet requires extensive revision and addition of missing information before it can be considered for publication. Please take the following comments into consideration:

1.  Line 51: "until now, only two pediatric studies exist." Two pediatric studies on which subject?

2. Line 70: what outcome?

3. Line 89-90: exclusion criteria are unclear: why did you choose to exclude only certain congenital malformations (why not congenital cardiac malformations?). What kind of illnesses are you referring to? Esophageal varices cannot be considered a congenital malformation of the upper gastrointestinal tract.

4. Line 98: which laboratory status?

5. Lines 146-147: "German Society for Ultrasound in Medicine recommendations". Please add an appropriate citation

6. Line 174: you applied a 95% confidence interval (CI), you cannot state that you have presented data as a 95% CI. Furthermore, values should be represented as mean ± standard deviation.

7. LD-pLT and DD-pLT have not been defined the first time they appear within the text.

8. Lines 196-197: please detail type of metabolic illnesses, origin of acute liver failure and type of congenital diseases.

9. Table 1: please represent data as mean (with decimals) ± standard deviation and gender as percentage. This is also valid for other data which has not been represented with standard deviations.

10. Line 210: Is this a mean value of the APP?

11. Please add to each p the value before the decimal (example: p<0.001)

12. Line 220: p lower than 0???

13. Table 2: there is no data clarifying how renal, cardiovascular or pulmonary dysfunction were defined. Chi square tests could be applied to this table.

14. Tables 3a and 3b should mention the dependent/independent variable based on which the multiple linear regression was applied.

15. Please add a headline in table 3c to the analyzed parameters.

16. Please clearly redefine the parameters in supplementary table 2. For example, what does spleen refer to? Supplementary table 1 could also benefit from inferential statistics.

17. Line 276: please add numeric data within the text.

18. Table 4: you could clarify in the headline of the last column that you have applied Mann-Whitney test for IAH vs ACS  comparison and provide data for each row.

19. Lines 308-309: I could not find this data in supplement table 2.

20. There is something wrong with each of the citations inserted in the discussion chapter. Please make sure to rectify this problem.

21. The discussion section should not represent a reiteration of the results obtained, but a comparison with other literature data (even those found in adult patients).

22. Please correct grammar mistakes and English language use errors.

23. Abstract: please rephrase the first sentence in order to make it more clear for the reader. Please rephrase lines 22-24, in order to state the objectives of the study. Please also rephrase the last line, to make it more appealing to the reader.

Round 2

Reviewer 2 Report

The article has undergone extensive revision and the authors did a great job in improving its quality. I feel that two of my comments have not been adressed properly:

1. Line 96 and previously comment 3: you cannot leave the term "diseases" when describing exclusion criteria, as this is highly unspecific. Please find a more appropriate syntagm.

2. Previous comment 5: please add this reference in a format which respects the journal's guidelines, not a pdf link.

Author Response

Again, very much thanks to the the reviewer for the very helpful and constructive feedback.

1.) We specified the exculsion criteria in the manuscript.

2.) We added a appropriate reference now.